# Genome-Wide Analysis of the FBA Subfamily of the Poplar F-Box Gene Family and Its Role under Drought Stress

**DOI:** 10.3390/ijms24054823

**Published:** 2023-03-02

**Authors:** Cong-Hua Feng, Meng-Xue Niu, Xiao Liu, Yu Bao, Shujing Liu, Meiying Liu, Fang He, Shuo Han, Chao Liu, Hou-Ling Wang, Weilun Yin, Yanyan Su, Xinli Xia

**Affiliations:** National Engineering Laboratory for Tree Breeding, College of Biological Sciences and Technology, Beijing Forestry University, Beijing 100083, China

**Keywords:** F-box protein, *P. trichocarpa*, *FBA* genes, drought stress

## Abstract

F-box proteins are important components of eukaryotic SCF E3 ubiquitin ligase complexes, which specifically determine protein substrate proteasomal degradation during plant growth and development, as well as biotic and abiotic stress. It has been found that the *FBA* (F-box associated) protein family is one of the largest subgroups of the widely prevalent F-box family and plays significant roles in plant development and stress response. However, the *FBA* gene family in poplar has not been systematically studied to date. In this study, a total of 337 F-box candidate genes were discovered based on the fourth-generation genome resequencing of *P. trichocarpa*. The domain analysis and classification of candidate genes revealed that 74 of these candidate genes belong to the *FBA* protein family. The poplar F-box genes have undergone multiple gene replication events, particularly in the *FBA* subfamily, and their evolution can be attributed to genome-wide duplication (WGD) and tandem duplication (TD). In addition, we investigated the *P. trichocarpa FBA* subfamily using the PlantGenIE database and quantitative real-time PCR (qRT-PCR); the results showed that they are expressed in the cambium, phloem and mature tissues, but rarely expressed in young leaves and flowers. Moreover, they are also widely involved in the drought stress response. At last, we selected and cloned *PtrFBA60* for physiological function analysis and found that it played an important role in coping with drought stress. Taken together, the family analysis of *FBA* genes in *P. trichocarpa* provides a new opportunity for the identification of *P. trichocarpa* candidate *FBA* genes and elucidation of their functions in growth, development and stress response, thus demonstrating their utility in the improvement of *P. trichocarpa*.

## 1. Introduction

In eukaryotes, the ubiquitin/26S proteasome system (UPS) is responsible for the selective degradation of most intracellular proteins [1]. There are three major enzymes involved in ubiquitin degradation, including the ubiquitin-activating enzyme (E1), ubiquitin-binding enzyme (E2), and ubiquitin-ligase (E3). E3 are divided into the following four categories based on their mechanism of action and subunit positions: the HETC structural domain type, RING-finger structural domain type, U-box structural domain type, and the SCF (Skp1-Cullin -F-box) domain type [2]. F-box proteins play a key role in recruiting substrates to UPS [3,4]. F-box proteins carry at least one 40–50 residue F-box/F-box-like domain at its N-terminus, which is responsible for binding to Skp1/Skp1-like proteins. At the same time, one or more conserved domains can also be found at the C-terminus, which are associated with substrate-specific recognition, including F-box-associated (FBA), kelch repeat and leucine-rich repeat (LRR) domains, etc. [5]. To date, F-box proteins have been discovered in almost all plants, and these F-box members form a larger protein family, of which the *FBA* subfamily is one of the largest subgroups [6]. According to previous studies, researchers found 94 F-box proteins containing FBA domains in *Triticum aestivum* [7], 46 in *Gossypium hirsutum* [8], 25 in *Cicer arietinum* [9], 14 in *Dioscorea esculenta* [10], 34 in *Juglans regia* [11], 278 in *Brassica rapa* [12], 17 in *Zea mays* [5], 196 in *Arabidopsis thaliana* [6], and 133 in *Malus domestica* [13].

A large number of F-box genes in plants are involved in the regulation of many biological processes, including hormone responses, lateral root formation, meristem formation, photomorphogenesis, senescence and stress response (abiotic and biotic processes) [9,14]. For example, the F-box protein TIR1 is an auxin receptor in *Arabidopsis thaliana*. Auxin-induced interaction between the Aux/IAA transcriptional repressor proteins and the ubiquitin–ligase complex SCF^TIR1^ mediates Aux/IAA degradation and auxin-regulated transcription [15]. The *Arabidopsis* F-box protein UFO is more likely to be a co-regulator that functions together with *LFY* in controlling organ-identity genes [16]. *EID1* possibly operates by targeting activated components of the phyA signaling pathway to ubiquitin-dependent proteolysis [17]. *TaFBA1*-OE plants improve drought tolerance by increasing antioxidant competence and decreasing ROS accumulation [18,19]. The information on plant F-box genes was mainly obtained from the research on *Arabidopsis*, rice and wheat, while the research on poplar F-box genes (*PtrFBXs*) is quite limited, and the research on F-box/FBA proteins is even more sparse. Hua et al. conducted a phylogenetic comparison of F-box proteins in 18 plants including poplar in 2011, and the results revealed the expansion, evolutionary selection, and functional relatedness of the F-box gene family. The results also indicate that the diversification of F-box genes may be achieved through genome drift [4]. In 2008, Yang et al. performed a comparative genomics analysis of the woody perennial plant poplar with the herbaceous annual plants *Arabidopsis thaliana* and rice, and elucidated the functional significance of this gene family [6]. Nonetheless, genome-wide analysis of *PtrFBXs* in poplar needs to be further studied.

Drought severely restricts plant growth and crop yields, and poplars are ideal materials for studying the response of trees to drought stress [20]. Some studies have been carried out on stress resistance genes in poplars. Previous studies have shown that plants have some intrinsic mechanisms to alleviate drought stress, including closing stomata, reducing transpiration, producing abscisic acid (ABA) and increasing surface wax. For example, *PeCHYR1* enhanced drought tolerance by promoting H_2_O_2_-mediated stomatal closure in poplar [21]. The overexpression of *PePYL4* led to higher water-use efficiency (WUE) and drought tolerance in poplar [22]. *PeABF3* improves drought tolerance by promoting ABA-induced stomatal closure by regulating the expression of *PeADF5* under drought conditions [23]. *PeSHN1* regulates both WUE and drought tolerance in poplar by modulating wax biosynthesis [24]. In recent years, whole genome sequencing of *P. trichocarpa* has been carried out for the fourth generation of resequencing, providing a valuable reference resource for the analysis of whole gene families, such as the PPO polyphenol oxidase gene family and the gibberellin-stimulated *Arabidopsis* (GASA) protein under abiotic stress [25,26]. However, the F-box family and its subfamily features have not been reported in poplar.

In this study, we aimed to identify and characterize the functions of *PtrFBAs* from the *P. trichocarpa* genome. We searched the *P. trichocarpa* genome using the HMMER website and identified 74 *PtrFBA* candidate genes. We predicted and analyzed the chromosomal localization, gene amplification, gene structure and upstream promoter cis-acting elements of the candidate *PtrFBA* genes. The chemical properties, subcellular localization, motifs and phylogenetic relationships of the proteins they encode were also predicted and analyzed. Focusing on identifying *PtrFBAs* associated with drought stress, the subcellular localization of the representative gene *PtrFBA60* and its germination rate and physiological phenotype under drought stress were analyzed. In this study, the *PtrFBA* gene was comprehensively analyzed by genomics, transcriptomics, and qRT-PCR for the first time, and the representative gene *PtrFBA60* was screened. We performed subcellular localization of *PtrFBA60* in tobacco and drought stress experiments in Arabidopsis thaliana, and analyzed the seed germination rate and physiological phenotype under drought stress to further study *PtrFBA’s* family properties and its possible role in drought tolerance in poplar.

## 2. Results

### 2.1. Genome-Wide Identification of PtrFBXs and PtrFBAs Families in P. trichocarpa

We searched the genome-wide protein database for the conserved F-box (PF00646) and FBA1(F-box associated 1) (PF07734) domains. The motif maps (Appendix A) and hmm model files of the two domains were downloaded; 459 sequences of the whole genome of *P. trichocarpa* were screened. Redundant sequences without F-box structural domains were then manually removed, resulting in a total of 337 F-box genes. These genes were named according to the order of corresponding chromosomal positions identified in the NCBI database. Different alternative splice forms from the same locus were assigned under the same gene names, such as *PtrFBA16* and *PtrFBA16.1* (Appendix A). The domain analysis showed that 101 out of 337 F-box genes contained only F-box domains (accounting for about 30%), 74 genes contained FBA domains (accounting for about 21%) and the other genes showed FBD, PP2, WD40, Kelch-type, LRR, FIST, NBD, DUF, AMN, TUB, RBD, PLN, PP1, or other domains (Figure 1). In different plants, the types and numbers of C-terminus domains are different, which indicates that the evolutionary history of different domains at the C-terminus may be more complex. Previous studies have found that the number of clans in the F-box family and the *FBA* subfamily is different in different plants. We performed an evolutionary developmental analysis of the *FBA* subgroup in *P. trichocarpa* and divided it into four subgroups (Figure 2a). We next compared the number of F-box family and *FBA* subfamily genes in ten species; the results showed that the gene number of *FBA* subfamilies and F-box family varied between species. The highest number of *FBA* genes was found in *Brassica rapa* with 278 genes, and the number of F-box genes in wheat reached 1013 (Figure 2b). F-box motifs and FBA domains usually contain about 50 amino acids, and several residues are conserved. We screened *PtrFBA60* as a representative gene from poplar’s *FBA* family. *Malus domestica* FBA protein *MdFBX3*, *Antirrhineum hispanicum* FBA protein *AhSLF4*, *Zea mays* FBA protein *FBX230.1*, *Arabidopsis thaliana* FBA protein *AtFOA2* and *Triticum aestivum* FBA protein *TaFBA1* for sequence alignment by using CLUSTALX (Appendix A). The results showed that they were conserved in F-box and FBA motifs.

We next analyzed the physicochemical properties of the 74 *PtrFBAs* genes in the en-tire *PtrFBAs* gene family. Except for *PtrFBA34*, which encodes about 191 amino acids, the *PtrFBAs* genes in *P. trichocarpa* generally encode 277–466 amino acids, with an average of 396 amino acids. The molecular weights of *PtrFBAs* proteins were both larger than 30 KDa, except for *PtrFBA34*. The theoretical pI of *PtrFBAs* was 4.74–9.51; 31 genes belonged to acidic proteins and 43 belonged to basic proteins. In addition, 30% of these FBA proteins were stable with the instability index <40. The overall hydrophilicity of the proteins is relatively strong (Appendix A). According to the prediction of the Plant-mPLoc database (http://www.csbio.sjtu.edu.cn/bioinf/plant-multi/, accessed on 15 July 2021), most of the *PtrFBAs* proteins are located in the nucleus, and a few in the membrane system.

### 2.2. Genomic Localization and Collinearity Analysis of PtrFBA Genes

The 74 *PtrFBA* genes were unevenly distributed on poplar chromosomes 1–19, with the largest distribution on chromosome 1 and none on chromosome 9 (Figure 3). *PtrFBA* repeat gene clusters were found on chromosomes 1, 3, 5, 6, 8, 10, 11, 12, 13, 15, 16, and 17 of *P. trichocarpa*. The largest repeat gene cluster had five genes, and is located on chromosome 6. To illustrate the expansion pattern of *PtrFBA* genes, we analyzed duplication events in the *P. trichocarpa* genome. *PtrFBAs* have undergone multiple repetitive events during evolution. *PtrFBA2* and *PtrFBA15*, *PtrFBA7* and *PtrFBA68*, *PtrFBA12* and *PtrFBA24*, *PtrFBA12* and *PtrFBA21*, *PtrFBA11* and *PtrFBA60*, *PtrFBA20* and *PtrFBA29*, *PtrFBA26* and *PtrFBA41*, *PtrFBA27* and *PtrFBA42*, *PtrFBA25* and *PtrFBA65*, *PtrFBX118* and *PtrFBA71* all form an entire genome duplication (WGD) pair, indicating that they all share a common ancestor with each other (Appendix A). The Ka/Ks values of *PtrFBA7, PtrFBA68, PtrFBA65* and *PtrFBA25* were all greater than 1, indicating that these genes had positive selection effects (Appendix A). The Ka/Ks values of the remaining three groups of genes are all less than 1, indicating that these genes have the effect of purification selection. In poplar, we found 19 genes that formed 10 segmental repeat events, which were distributed on 11 chromosomes, with two additional tandem repeat regions on chromosomes 6 and 10 (Figure 4). Next, we constructed comparative syntenic maps of the F-box gene family of P. trichocarpa and four typical plants (Arabidopsis, wheat, maize and apple) (Appendix A). The syntenic maps showed that there were 129 pairs of *FBX* homologous genes between *P. trichocarpa* and *Arabidopsis thaliana*, 48 pairs of *FBX* homologous genes with maize, 124 pairs of FBX homologous genes with wheat, and 172 pairs of *FBX* homologous genes with apple. We then constructed comparative syntenic maps of the *FBA* gene families of *P. trichocarpa* and four species (Figure 5); it was found that *P. trichocarpa* has 8 pairs of *FBA* homologous genes with *Arabidopsis*, 1 pair of *FBA* homologous genes with maize, no *FBA* homologous genes with wheat, but 26 pairs of *FBA* homologous genes with apple. In addition, it was found that some *PtrFBA* genes are related to at least two synonymous gene pairs, especially *FBA* genes between poplar and apple, indicating that these genes may play an important role in the evolution of plants. The results of this study show that *PtrFBAs* have fewer homologs in wheat, *Arabidopsis* and maize and more homologs in the woody plant apple. This suggests that multiple *FBA* homologs may have been formed in woody plants during the differentiation process.

### 2.3. Gene Structure and Motif Identification of PtrFBA Gene Family

To more clearly explore the evolutionary relationship between different members of the *PtrFBA* gene family, their gene structures, conserved protein motifs, and conserved structures were evaluated (Figure 6b). Most *PtrFBA* genes (about three fifths of the genes) have no introns, six genes have two introns (*PtrFBA19/24/26/27/37/53*) and twenty-one genes have one intron. Moreover, 15 genes did not contain UTR, and 8 genes contained neither introns nor UTRs (*PtrFBA5/16.1/23/43/46/47/56/73*). At the same time, 10 conserved motifs were found in the *PtrFBA* protein sequence using MEME (Figure 6a), of which 2 motifs (1 and 8) were associated with the N-terminal F-box domain and 6 motifs (2, 5, 6, 9, 4, and 10) were associated with the C-terminal FBA domain (Appendix A).

### 2.4. PtrFBAs cis-Element Analysis

To determine the expression pattern of *PtrFBAs*, the sequence upstream of the *PtrFBA* family promoter (~2000 bp) was extracted from the *P. trichocarpa* genomic DNA sequence. The cis-elements of the *PtrFBA* promoter were analyzed using the PlantCARE database (Figure 7). We visualized the promoter elements in promoter positions and species (Appendix A). The specific functions of these elements (cis-elements) are annotated (Appendix A). These elements are involved in abiotic and biotic stresses, phytohormonal responses, and plant growth and development. Elements of the *PtrFBA* promoter are mainly involved in abiotic and biotic stresses, as well as phytohormone response factors. Stress-related cis-elements (Myb, Myc, ARE and STRE) were extracted, suggesting that *PtrFBAs* may play a key role in coping with adverse conditions. In addition, some *PtrFBA* promoters are enriched in ABRE (involved in the ABA response), such as *PtrFBA42/PtrFBA61/PtrFBA/65*, and these genes may be responsive to ABA hormones. *PtrFBA22/PtrFBA37* have more LTR elements (involved in low-temperature stress responses) in their promoters, suggesting that these genes may be responsive to low-temperature induction.

### 2.5. Transcriptome Analysis of PtrFBA Gene in P. trichocarpa

To further investigate the role of *PtrFBA* genes in growth and development, we examined the tissue expression profiles of 30 *PtrFBA* genes from transcriptome data (Appendix A). We created a heat map of the samples and genes clustered in both directions to explore the expression characteristics of *PtrFBA* genes in 15 poplar tissues (Figure 8a). Most of the genes were highly expressed in mature sites, while a few were highly expressed in young sites (e.g., *PtrFBA1/4/6*). Some genes were highly expressed in phloem (e.g., *PtrFBA49/50/55/72*), which may be related to the growth and development of poplar. We downloaded transcriptomic datasets to visualize the expression of 30 *PtrFBAs* genes after drought, beetle and mechanical damage (Appendix A) and performed transcriptomic data analysis and visualization to analyze the effects of *PtrFBA* genes on stress response in poplar (Figure 8b). Most of the genes in the *PtrFBA* gene family are sensitive to drought stress and their expression is up-regulated under drought stress. Some genes were up-regulated under leaves beetle damage stress (e.g., *PtrFBA11/26/65*), and some were up-regulated under leaves mechanical damage stress (e.g., *PtrFBA26/40/42*). *PtrFBA26, PtrFBA42* and *PtrFBA65* were up-regulated under both types of stress, indicating that these genes may respond to leaf injury stress.

### 2.6. Expression Patterns of Some PtrFBAs under Different Treatments

To further explore the function of *PtrFBAs*, we investigated the response of some *PtrFBAs* to drought stress by qRT-PCR in P. trichocarpa leaves (Figure 9). Three different gradients were set according to soil moisture content. *PtrFBA15/45/60/72* were up-regulated under drought stress, while *PtrFBA6/11/13/36/50* were down-regulated, and *PtrFBA61* showed no significant change. We also found that *PtrFBA60* was significantly sensitive to ABA (Appendix A), and weakly sensitive to ethylene and NaCl (Appendix A).

### 2.7. Drought Treatment of Transgenic Arabidopsis

We selected the *PtrFBA60* gene for follow-up experiments. To determine the subcellular localization of *PtrFBA60*, the Super:: *PtrFBA60*-GFP (green fluorescent protein) fusion protein was transiently transfected into tobacco leaves. It was found to be likely located in the cytoplasm; however, it was not uniformly distributed in the cytosol, but rather formed punctate aggregates (Figure 10a). The *PtrFBA60* aggregates were found to correspond to P-bodies by co-localization experiments between *PtrFBA60*-GFP and the P-body markers AtTZF1-mCherry and AtAGO-mCherry (Figure 10b). The results indicate that *PtrFBA60* is localized in the P-bodies [27,28]. Since drought stress can induce the expression of *PtrFBA60* (Figure 9), we hypothesized that *PtrFBA60* plays a vital role in the drought response. The seed germination rate experiments showed that there was no significant difference in the germination rate of the *OE-PtrFBA60*, WT, and mutants *fba60* on 1/2 MS medium, but the germination rate of *OE-PtrFBA60* was significantly higher for the 200 mM mannitol cultured medium (Appendix A). The *OE-PtrFBA60,* mutant *fba60*, and WT Arabidopsis were cultured under the same conditions and then subjected to drought treatment. After 20 days, the leaves of mutant *fba60* were severely wilted and the leaves of WT turned yellow and wilted, while those of *OE-PtrFBA60* lines remained green and turgid. Furthermore, *OE-PtrFBA60* trangenic plants recovered normally after being re*watered*, while the mutant and WT Arabidopsis plants did not fully recover (Figure 11a). The survival rate of *OE-PtrFBA60* was 85.8%, while the survival rate of WT and mutant *fba60* was 51.6% and 36.6%, respectively (Figure 11b). The REC (relative electrical conductance) characterizes the degree of damage to plant cell membranes and the REC of *OE-PtrFBA60*, mutant *fba60*, and WT both increased after drought treatment, but the values for fba60 and WT were three times larger than those for *OE-PtrFBA60* (Figure 11c). These results indicated that *OE-PtrFBA60* transgenic Arabidopsis had better drought stress tolerance.

## 3. Discussion

As a component of the SCF E3 ligase complex, the F-box protein family plays an im-portant role in plant growth, development and stress resistance [29,30,31,32,33,34]. The *FBA* subfamily is a ubiquitous and abundant subfamily in the F-box family, consisting of an N-terminal F-box conserved domain and a C-terminal FBA conserved domain [18,19]. The *FBA* gene family is related to many plant physiological activities, such as plant growth and development, hormone signal transduction, abiotic and biotic stresses and plays a significant role [3,15,17,35,36,37]. In many plants (wheat, maize, *Arabidopsis*, etc.), the biological function and bioinformatic analysis of the *FBA* gene family has been widely reported, but this is not the case for the woody plant poplar [6]. In this study, 74 candidate *FBA* genes were identified in *P. trichocarpa*, and bioinformatics and expression patterns of these genes were analyzed. As is the case with maize, apple and *Arabidopsis*, the *FBA* genes in *P. trichocarpa* are unevenly distributed, and distributed on several chromosomes (Figure 3). These 74 proteins are all high-molecular-weight proteins, except *PtrFBA34*. The experimental results show that in agreement with other species, the predicted subcellular localization varies, but most are located in the nucleus. However, the subcellular localization of our target gene, *PtrFBA60*, showed that *PtrFBA60* was localized in the P-body (Figure 10) [27,28]. Due to the gap between their occurrence positions and distances, the *FBA* gene families were discretely distributed in the phylogenetic tree, and were mainly divided into four clusters (Figure 2). The origin and evolution of the *PtrFBAs* genes were also further determined.

The increase in gene family members and the mechanism of genome evolution mainly depend on gene duplication events, including tandem duplication and segmental duplication [38]. In the present study, 74 *PtrFBAs* genes were unevenly distributed on 19 poplar chromosomes, and this phenomenon of uneven distribution of chromosomes suggested that this change occurred before species differentiation (Figure 3). A total of 10 pairs of whole genome duplicate genes and 12 tandem duplicate genomes were detected in poplar. This indicates that both tandem repeat sequences and whole genome repeats are involved in the evolution of poplar *PtrFBAs* genes (Figure 4). Previous gene family studies have suggested that tandemly repeated genes may have similar functions and expression patterns. Most *PtrFBAs* have no introns, while some have up to two introns (Figure 6). In order to quickly respond to stress, organisms need to stimulate the expression of genes, and gene structures with few or no introns contribute to the rapid expression of mRNA [39,40]. Genes such as *PtrFBA3/4/7* lack introns but contain UTRs, so they can be transcribed faster to form mRNA.

A large number of genes for FBAs were present in each species (Figure 2b). Covariance analysis with four different species revealed that the F-box gene family and *FBA* gene family of poplar are more closely related and homologous to the woody plant apple than other species (Appendix A, Figure 5) [11,13,31,41,42,43,44,45]. This suggests that considerable FBA divergence and doubling may have occurred during the evolution of perennial woody plants.

The expression patterns of *FBA* genes in various tissues have been confirmed in many species [10]. Due to differences in the number of *FBA* genes in different species, there is no uniform *FBA* gene expression profile in plants. According to the RNA-seq data of poplar, some genes were preferentially expressed in the phloem and shoots (Figure 8a), indicating that these genes are essential for plant growth [3]. Some genes are highly expressed throughout the growth and development process, indicating that the *FBA* gene family plays an important role in the entire plant growth and development process [14,18,32,33,35].

The expression of the *FBA* gene family in plants is closely related to stress and adversity, mainly in response to drought, insects and mechanical damage [5,18]. Transcriptome data showed that drought significantly induced some *FBA* genes in poplar (*PtrFBA2/15/60*) (Figure 8b), which was verified by quantitative reverse transcription polymerase chain reaction (qRT-PCR) (Figure 9). By analyzing the promoters of the *FBA* gene family, we found that there are many different elements in their promoters, especially ABRE and ARE, which are involved in biotic and abiotic stresses (Figure 7) [14,46,47,48]. Furthermore, ABRE elements play an important role in response to abiotic stress, which ensures that *FBA* genes can be rapidly induced under stress [49]. In order to verify whether the *FBA* gene can provide plants with the function of drought resistance, we overexpressed the *PtrFBA60* gene in the model plant *Arabidopsis thaliana* and purchased *Arabidopsis thaliana* seeds with mutant *fba60*. It was found that overexpression of *PtrFBA60* significantly improved the drought resistance of *Arabidopsis* (Figure 11), and the expression of *PtrFBA60* also significantly increased under ABA treatment, which may improve plant drought resistance through the ABA pathway (Appendix A) [19,46,50,51,52]. In summary, plants can adapt to various complex environments by regulating their expression level of *FBA* genes.

## 4. Materials and Methods

### 4.1. Identification and Evolutionary Analysis of the FBA Family of Populus Trichocarpa

The identification and analysis of the *FBA* gene family were based on the whole genome of *P. trichocarpa*. Relevant poplar genomic and protein data were downloaded from the Phytozome13.0 database (https://phytozome-next.jgi.doe.gov/, accessed on 12 July 2022). We downloaded the Hidden Markov model of the FBA1 structural domain (PF07734) and F-box structural domain (PF00646) from the Pfam database (http://pfam.xfam.org/, accessed on 30 January 2023), and searched the genome of *P. trichocarpa* using the Simple HMM Search program in TBtools software (TBtools_windows-x64_1_098685) and predicted the IDs of the *PtrFBAs* gene family. These sequences were further confirmed by the Protein BLAST function of the NCBI database (https://blast.ncbi.nlm.nih.gov/Blast.cgi, accessed on 30 January 2023) [53,54]. The *P. trichocarpa 4.1* genome, CDS, transcripts, polypeptides and 2000 bp upstream of the promoter region of the translation initiation site (ATG) were extracted from the Phytozome 13.0 database. The amino acid sequences of *PtrFBX* and *PtrFBA* target sequences were analyzed by ClustalX. In addition, using the neighbor-joining (NJ) method parameter on MEGA7.0 with pairwise deletion and 1000 bootstrap replicates, multiple alignments were applied to construct a phylogenetic tree among PtrFBAs [54,55]. PtrFBAs and *PtrFBXs* are named according to their positions on the scaffold and chromosomes, respectively.

### 4.2. Sequence Alignment, Physicochemical Properties, Subcellular Localization Prediction and Gene Positions

The amino acid sequences of each species were obtained from the NCBI database, the ClustalW (https://www.genome.jp/tools-bin/clustalw, accessed on 30 January 2023) website was used, the Clustal algorithm was used for sequence alignment, and the ENDscript/ESPript (https://espript.ibcp.fr/ESPript/cgi-bin/ESPript.cgi, accessed on 30 January 2023) website was opened to map the sequence alignment results. The physicochemical properties of the *PtrFBAs* family were analyzed using the TBtools Protein Paramter Calc function. Subcellular localization was predicted by the site Plant-mPLoc (http://www.csbio.sjtu.edu.cn/bioinf/plant-multi/, accessed on 30 January 2023) [56,57] and the predicted subcellular localization of the *PtrFBAs* family was obtained by entering their amino acid sequences into the website. Gene positions and chromosome sizes of *PtrFBAs* and *PtrFBXs* were obtained from the NCBI database and visualized by TBtools (South China Agricultural University, Guangzhou, China) [58].

### 4.3. Identification of Gene Duplication Events and Collinearity Relationship

The *PtrFBX* and *PtrFBA* duplication events in the poplar genome were evaluated using multiple collinear scanning toolkits (MCScanX) [59]. The poplar genome and gff3 files were applied to investigate the putative duplication events using TBtools with one-step MCScanX with default parameters. In addition, the ratio of non-synonymous (Ka) and synonymous (Ks) substitutions in *PtrFBA* gene pairs were measured to evaluate the selection pressure of the *PtrFBAs* evolutionary process. To further evaluate the collinearity relationship among poplar, ***Arabidopsis***, apple, maize and wheat, the MCScanX with default parameters was also applied to identify the putative orthologs. In a similar manner to intra-poplar collinearity, the genome and gff3 files of poplar, ***Arabidopsis***, apple, maize and wheat were aligned to achieve the three important files, including the control file (ctl), gff and collinearity formats. We manually removed unnamed chloroplasts and mitochondrial chromosomes from the ctl files and reordered them, and finally visualized them with TBtools.

### 4.4. Analysis of Conserved Motifs and Gene Structures

We uploaded gff3 annotation files that contained gene information to the TBtools (https://github.com/CJ-Chen/TBtools, accessed on 30 January 2023) software (TBtools_windows-x64_1_098685)to reveal gene conserved motifs and gene structures. In addition, the *PtrFBA* protein sequences were submitted to Motif’s Multiple Expectation Maximization (MEME) web version (https://meme-suite.org/meme/tools/meme, accessed on 30 January 2023) [60]. In addition, gff3 files and patterns stored in the Extensible Markup Language (XML) format were uploaded to the TBtools software for displaying conserved motifs and gene structures.

### 4.5. Analysis of Cis-Regulatory Elements

The cis-acting elements of promoters were predicted by PlantCARE (http://bioinformatics.psb.ugent.be/webtools/plantcare/html/, accessed on 30 January 2023), and classified and analyzed according to the literature, and the results were visualized by EXCEL and TBtools.

### 4.6. Transcriptome Analysis and Visualization of the PtrFBA Family

To analyze the expression profile of the *PtrFBA* family, we searched and downloaded the transcriptome data of 30 *PtrFBAs* genes from the PopGenIE (https://plantgenie.org/, accessed on 30 January 2023) public database [61,62]. In this database, expression data were collected for three abiotic stresses and one biotic stress, as well as expression data for 15 different plant tissues. Heat maps of the relative expression quantification of *PtrFBAs* were created using TBtools, and Row Scale was chosen for homogenization, while all the other parameters were set as default parameters.

### 4.7. Plant Materials and Sample Collections

It was modified to: Annual poplar seedlings were planted in the nursery of Beijing Forestry University, with two poplar seedlings planted in each pot. *P. trichocarpa* seedlings were grown for about 3 to 4 months under the control of a greenhouse environment. Under different abiotic stresses, 5 potted seedlings with the same growth conditions were selected for each abiotic stress treatment. All collected samples were immediately frozen in liquid nitrogen and stored in a −80 °C freezer. After returning to the laboratory, RNA extraction (kit from Aidlab biotechnologies CO.LTD., Beijing, China) and reverse transcription (kit from TIANGEN BIOTECH CO.LTD.) were performed.

*Arabidopsis* ecotype Colombia (Col-0) was used in this study. The *fba60* (SALK_019628.33.80.x) mutant was obtained from the *Arabidopsis* biological resource center ABRC (ABRC (osu.edu)). *fba60* is a T-DNA insertional mutant. Plants were grown in soil or 1/2 MS medium and cultured in a constant temperature incubator (16 h light/8 h dark cycle, 22 °C).

### 4.8. Quantitative Real-Time (qRT-PCR) Analysis

qRT-PCR analysis-specific primers were designed and tested by NCBI Primer BLAST (https://www.ncbi.nlm.nih.gov/tools/primerblast/index.cgi?LINK_LOC=BlastHome, accessed on 30 January 2023). Real-time PCR was run with the CFX96 Touch™ instrument (Bio-Rad, Hercules, CA, USA) to detect the chemical SYBR Green. The reaction system established was as follows: 10 µL of 2× SuperReal Premix Plus (TIANGEN), 0.5 µL of each forward and reverse primer, 1 µL of diluted cDNA template (100 ng/µL), and finally, RNase-free ddH_2_O was added until the total was 20 µL. The reaction procedures were as follows: 95 °C for 15 min, 45 cycles of 10 s at 95 °C and 30 s at 60 °C. We used two internal reference genes to conduct the experiment. The selection of internal reference genes and analysis of the experimental results were based on the research of Wang et al. [63,64]. SPSS (IBM, Armonk, NY, USA) was used for statistical analysis, and the LSD test was used to calculate the *p* value. All the primers used are listed in Appendix A.

### 4.9. Subcellular Localization of PtrFBA60 Proteins

To determine the subcellular localization of *PtrFBA60* in plant cells, the full-length CDS of *PtrFBA60* was cloned into the pSuper-1300 vector, generating a *PtrFBA60*-GFP fusion protein, and then introduced into *A. tumefaciens GV3101*. A plasmid that contained GFP alone served as a control. Similarly, the full-length CDS of the P-body markers gene *AtAGO* and *AtTZF1* was cloned into the pSuper-1300 vector to generate *AtAGO*-mCherry and *AtTZF1*-mCherry fusion protein, and then introduced into *A. tumefaciens GV3101*. Transient expression assay in tobacco leaf epidermal cells was performed as previously described [65]. The GFP or mCherry signal was detected by laser confocal microscopy SP8 after two days of infestation (Leica Microsystems TCS SP8, Wetzlar, Germany).

### 4.10. Cultivation of Arabidopsis, Construction of Vector and Transformation of Arabidopsis and Identification of Transgenic Arabidopsis

The *PtrFBA60* gene was cloned using *P.trichocarpa* as a template, and *PtrFBA60* was inserted into the pBI121-GUS vector with *XbaI* and *XmaI* restriction sites. After fusion with the GUS in the pBI121 vector, the *OE-PtrFBA60* vector was obtained and transformed into *GV3101 Agrobacterium*. The inflorescence method was used to transform *Arabidopsis thaliana*. We chose the activated monoclonal positive Agrobacterium containing the target gene and placed it into 10 mL fresh YEB medium (containing 100 mg/L Kan, 50 mg/L Rif), shaken overnight in a constant temperature incubator at 28 °C and 200 rpm cultivated for 12 h. The overnight cultured bacterial solution was added to 100 mL of YEB culture solution (containing 100 mg/L Kan, 50 mg/L Rif), shaken again to achieve the OD600 = 0.8 and centrifuged at 5000× *g* rpm for 10 min. The supernatant was discarded and we used the resuspension solution (containing 2.18 g 1/2 MS, 0.3–0.4% foaming agent, 50 g sucrose; 0.5 g MES per liter), resuspending the agrobacterium until the OD600 was 0.8. *Arabidopsis thaliana* inflorescence was invaded in the infestation solution for 2–3 min.After completion, the solution was incubated in the dark for 24 h. The second infection experiment was performed after 3–4 days. For the histochemical GUS analysis, the prepared material was immersed in GUS staining solution and incubated overnight at 37 °C.

### 4.11. Determination of Physiological Indicators Such as Drought Treatment and Germination Rate of Arabidopsis Thaliana

Wild-type and transgenic *Arabidopsis* seeds were sown simultaneously on 1/2 MS solid medium at a mannitol concentration of 0 or 250 mmol/L. Fifty-six seeds of each line were sown equidistantly on each medium, and three technical replicates were made. After 2 days of dark culture, the cells were placed in a constant temperature incubator at 22 °C for 16 h in the light, 8 h in the dark, and the germination rate was recorded every 12 h. Wild-type and transgenic *Arabidopsis* seeds were also sown simultaneously in soil and grown in a climate-controlled chamber with a light intensity of 180 μ mol/m^2^/s, a light duration of 14 h/dark duration of 10 h, and a temperature of 25 °C. *Arabidopsis* were subjected to natural drought treatment for 20 days after one month of growth. Wild-type and mutant *Arabidopsis* leaves were photographed after water loss and yellowing, and finally treated with rehydration for three days and counted for viability and REC.

### 4.12. Statistical Analyses

Statistical analyses were conducted using Microsoft Excel 2016 (Microsoft Corporation, Redmond, WA, USA) and SPSS (version 25, IBM Corporation, Armonk, NY, USA). One-way ANOVA with the LSD multiple comparisons test (* *p* < 0.05, ** *p* < 0.01, *** *p* < 0.001.) was performed for the gene relative expression. Before applying the ANOVA test, the data were tested for normality and homogeneity of variance. Student‘s *t* test (* *p* < 0.05; ** *p* < 0.01) was performed for the survival rates and REC.

## 5. Conclusions

We believe that both tandem and WGD repeats were involved in the evolution of *PtrFBAs* genes in poplar, and that perennial woody plants may have undergone considerable *FBA* gene differentiation and doubling during evolution. Moreover, *PtrFBA* genes play a significant role in the process of plant growth and development, and also play a regulatory role in adversity stress. Among them, *PtrFBA60* plays a positive regulatory role in plant drought resistance, which may allow plants to acquire drought resistance through the ABA pathway. Future experiments will aim to investigate the nature of these results from a genetic, biochemical and physiological perspective.

## Figures and Tables

**Figure 1 ijms-24-04823-f001:**
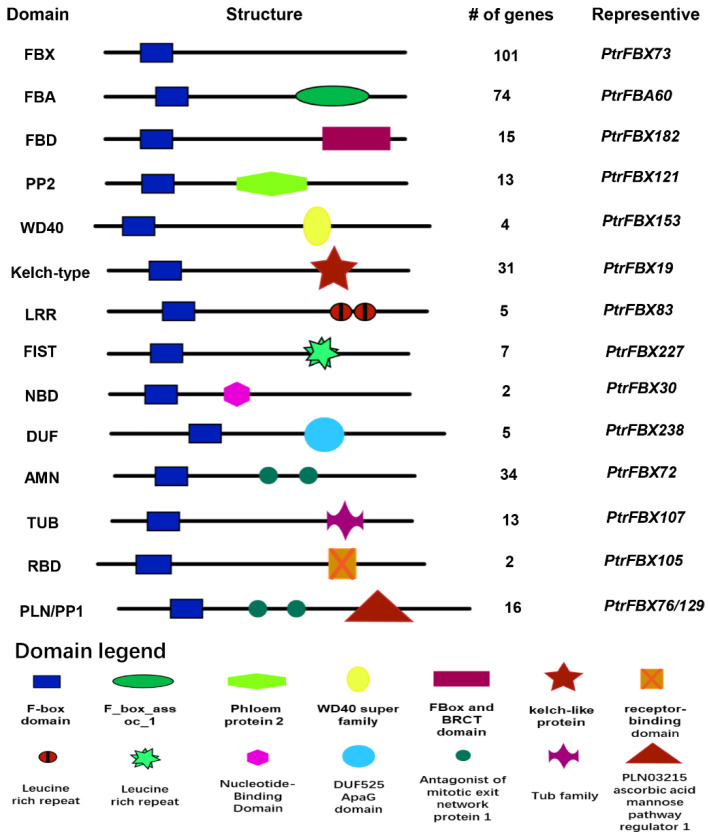
A map of *P. trichocarpa* canonical F-box proteins with structural and positional information of the C-terminal interacting domains; shown on the left is the type of the C-terminal domain and on the right are the number and representative genes predicted to contain F-box proteins of these domains. Below are the comments for each structure field.

**Figure 2 ijms-24-04823-f002:**
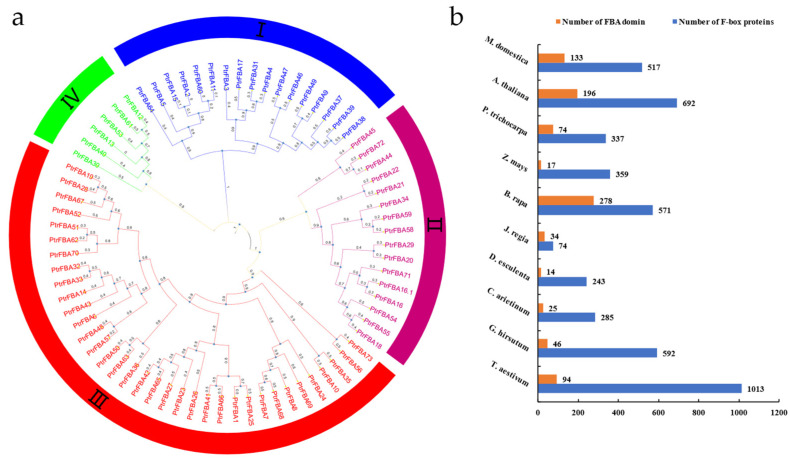
Research of *FBA* gene family in *P. trichocarpa* and multiple species. (**a**) Evolution and phylogenetic analysis of the *PtrFBAs* family in *Populus trichocarpa*. The FBA protein family in *P. trichocarpa* have been divided into four subgroups according to branch color. (**b**) Number of F-box gene families in ten species and number of *FBA* gene subfamilies among them. *M. domestica*, *Malus domestica*; *A. thaliana*, *Arabidopsis thaliana*; *P. trichocarpa*, *Populus trichocarpa*; *Z. mays*, *Zea mays*; *B. rapa*, *Brassica rapa; J. regia*, *Juglans regia*; *D. esculenta*, *Dioscorea esculenta*; *C. arietinum*, *Cicer arietinum*; *G. hirsutum*, *Gossypium hirsutum*; *T. aestivum*, *Triticum aestivum*.

**Figure 3 ijms-24-04823-f003:**
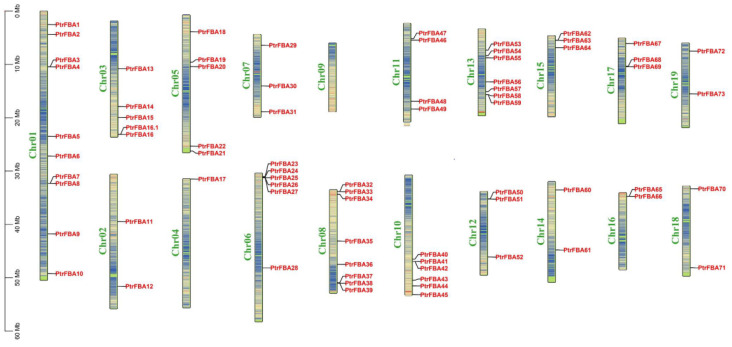
Localization of 74 FBA proteins on 19 *P. trichocarpa* chromosomes. The scale on the left is in megabytes. The serial number of the chromosomes is shown to the left of each image. The gene names to the right of each chromosome correspond to the approximate location of each *PtrFBA* gene. Chromosome colors represent gene abundance.

**Figure 4 ijms-24-04823-f004:**
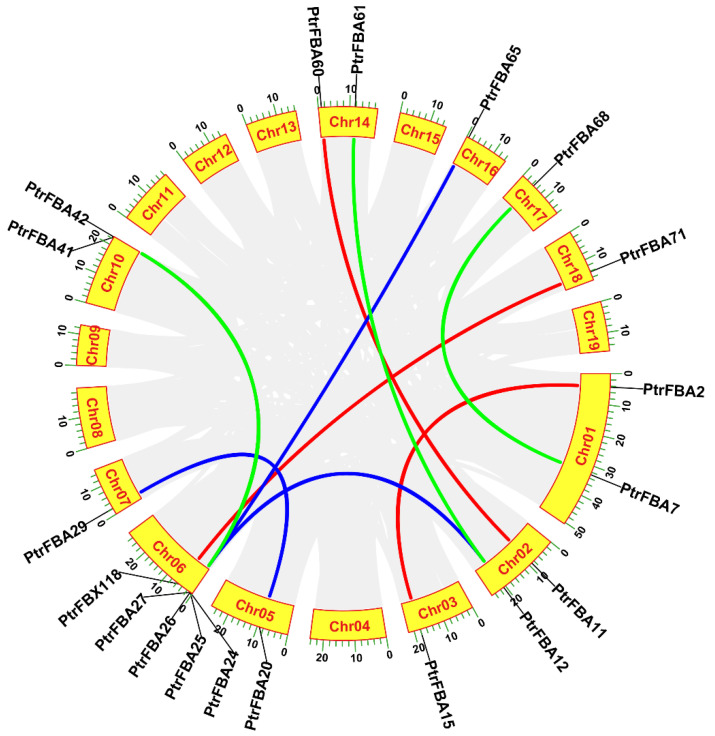
Evolutionary relationship analysis of the *PtrFBAs* gene family. Evolutionary analysis of the *PtrFBAs* gene family in *P. trichocarpa*, with different sizes of fan-shaped rings and different sizes of chromosomes. The gray lines in the background represent collinear blocks of the genome and the colored lines represent paralog *PtrFBA* gene pairs.

**Figure 5 ijms-24-04823-f005:**
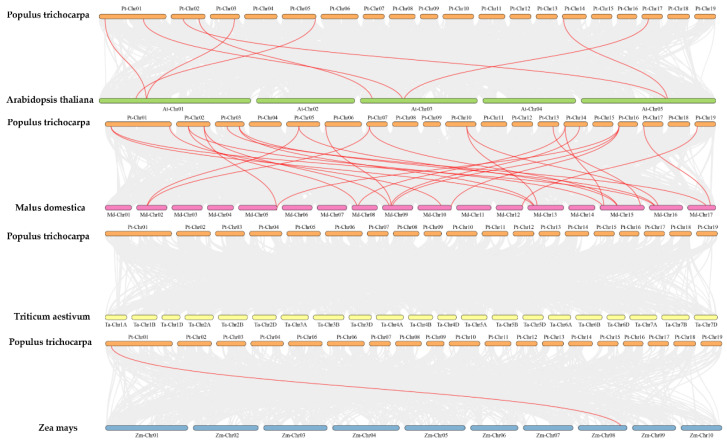
Collinear analysis of poplar *PtrFBAs* gene family with 4 typical plants (*Arabidopsis*, wheat, maize and apple). Gray lines in the background represent collinear blocks of *P. trichocarpa* and other species genomes, while red lines emphasize collinear *PtrFBAs* gene pairs.

**Figure 6 ijms-24-04823-f006:**
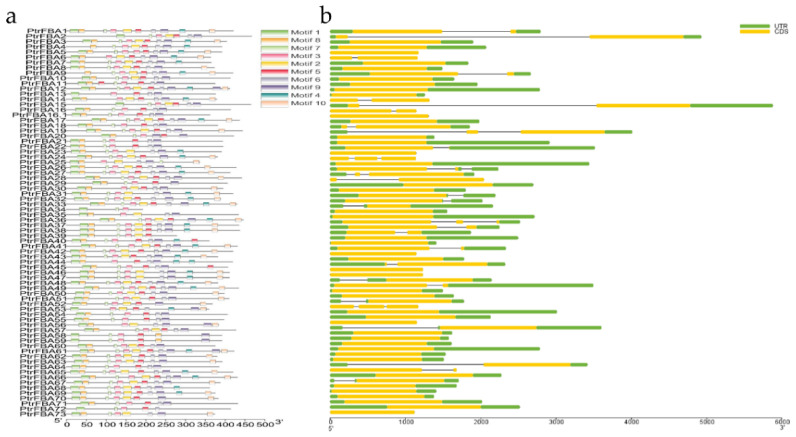
Conserved protein structure and gene structure analysis of the *PtrFBA* gene family of *P. trichocarpa.* (**a**) Motif analysis of the *PtrFBA* gene family of *P. trichocarpa*. Individual motifs are displayed in square blocks of different colors, and the type is numbered on the square blocks. (**b**) Gene structure analysis of the *PtrFBA* gene family of *P. trichocarpa.* Upstream and downstream non-coding regions are indicated by green rounded rectangles, exons are indicated by yellow rounded rectangles, and introns are indicated by gray lines.

**Figure 7 ijms-24-04823-f007:**
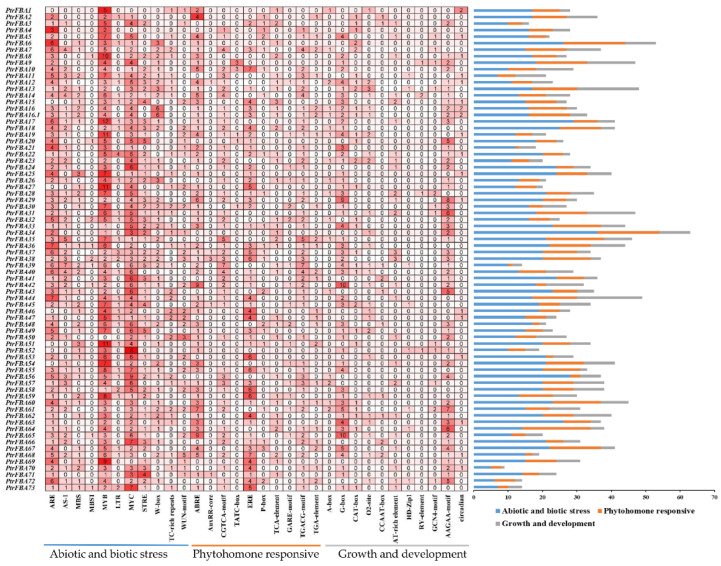
The cis-acting element of the *PtFBA* gene. Numbers and red gradient indicate the number of cis-acting elements. Color-coded histograms indicate the number of cis-acting elements for each type of gene, which are divided into the following three categories by functional factors: phytohormone responsive, abiotic and biotic stress, plant growth and development.

**Figure 8 ijms-24-04823-f008:**
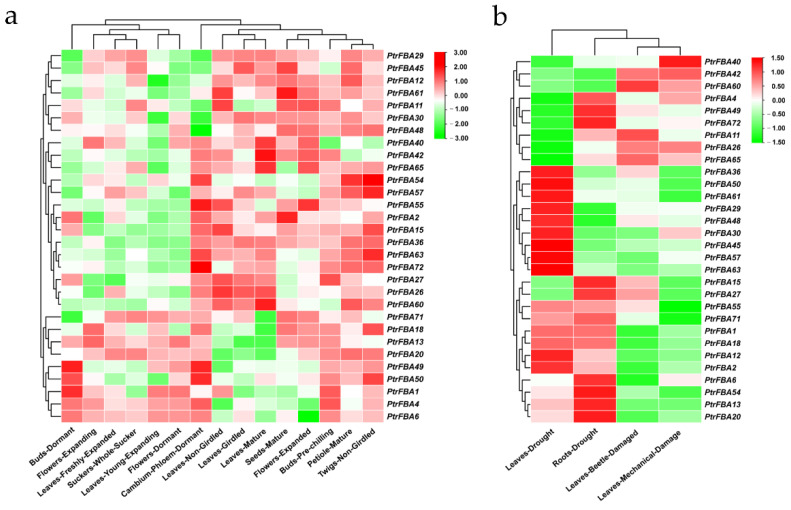
Expression profiles of *PtrFBA* genes under developmental and stress conditions. (**a**) The expression levels of 30 *PtrFBAs* genes in different tissues at different developmental stages are plotted based on transcriptome data. (**b**) A heat map of gene expression levels of 30 *PtrFBAs* following drought, beetle and mechanical injury. The color bar represents the range of maximum and minimum values of relative expressions in the heatmap. Red, white and green represent high, medium and low expression levels, respectively.

**Figure 9 ijms-24-04823-f009:**
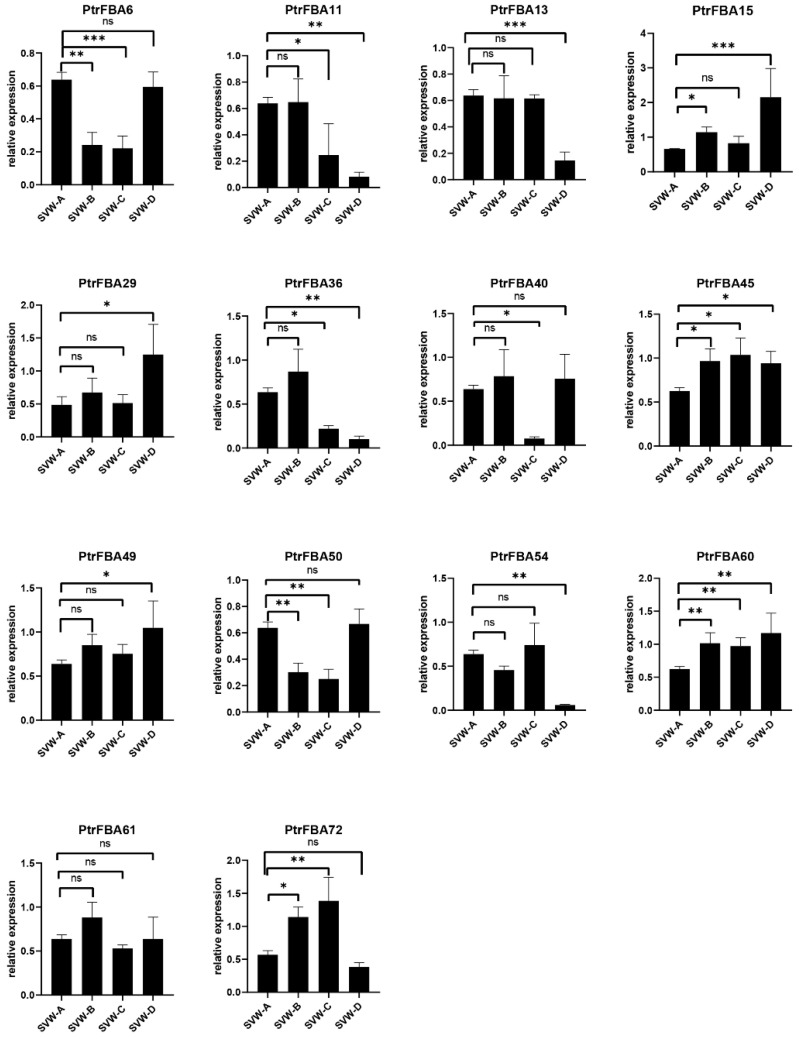
Expression analysis of *PtrFBAs* genes in response to drought stress. SVW-A, SVW-B, SVW-C, and SVW-D represent four drought treatments with different soil volumetric water content (soil-VWC): control (43 ± 1% soil-VWC), mild drought (33 ± 1% soil-VWC), moderate drought (23 ± 1% soil-VWC) and severe drought (13 ± 1% soil-VWC), respectively. One-way ANOVA with LSD multiple comparison, n = 3. ns; * *p* < 0.05; ** *p* < 0.01, *** *p* < 0.001.

**Figure 10 ijms-24-04823-f010:**
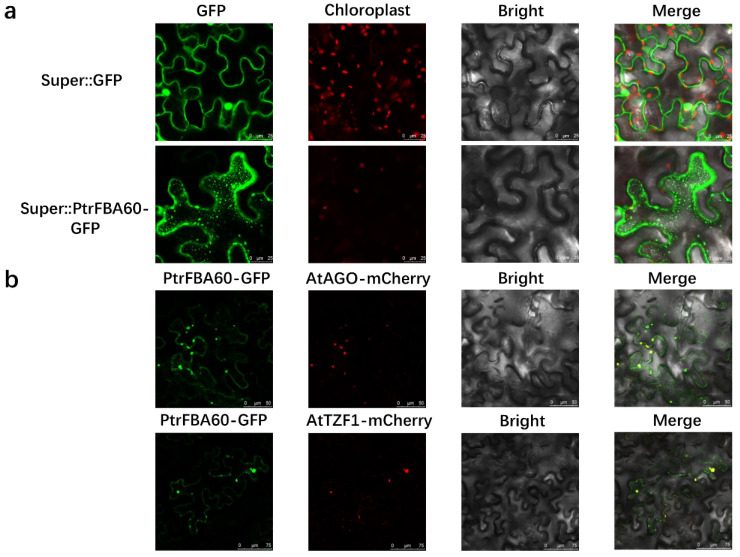
The location of PtrFBA60 in P-Bodies **of** tobacco. (**a**) The Super::GFP and Super::*PtrFBA60*-GFP fusion proteins were transiently expressed in tobacco. (**b**) *PtrFBA60*-GFP and AtAGO-mCherry, AtTZF1-mCherry co-location in tobacco and *PtrFBA60* colocalized with PB (P-body) markers AtAGO1 and AtTZF1.

**Figure 11 ijms-24-04823-f011:**
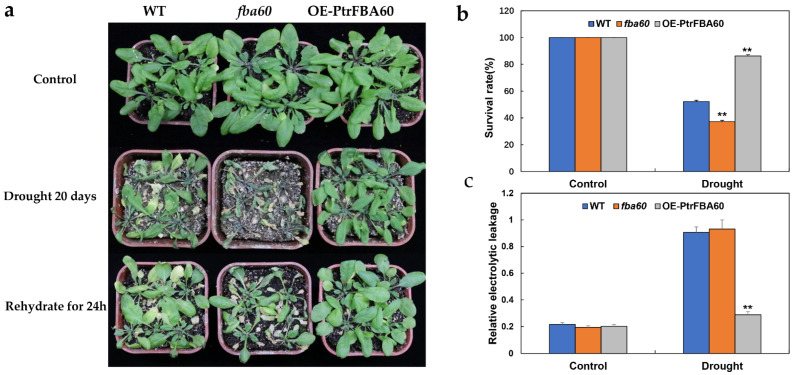
Drought stress treatment in transgenic Arabidopsis thaliana. (**a**) The phenotypes of wild-type Arabidopsis thaliana (WT), mutant *fba60* and *OE-PtrFBA60* after drought and being rewatered. (**b**) The survival rate of each line after drought treatment. (**c**) REC of each line after drought treatment. Values shown are averages ± SE from three replicates (n = 50 in each replicate). ** *p* < 0.01 (Student’s *t* test) with respect to the wild type values in the same experimental conditions.

## Data Availability

All the data that support the findings of this study are available in the paper and its Appendix A published online.

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
