# Peer review of "Genome-Wide Analysis of the FBA Subfamily of the Poplar F-Box Gene Family and Its Role under Drought Stress"

_ijms, 2023, doi:10.3390/ijms24054823_

Round 1

Reviewer 1 Report (Previous Reviewer 1)

The paper focus on the FBA subfamily of F-box proteins of poplar. In silico genome wide analysis to evaluate size and features of the family is followed by analysis of expression (by transcriptomics and real-time) of some representative FBAs in stress conditions. Finally, results of under- or over- expression in transgenic plants of a selected interesting component with a regulatory role in plant drought resistance is reported.

The paper is generally well structured and I believe that it has been significantly improved after first revision. It still needs some minor revisions

See attached file for corrections

Author Response

-Dear Reviewer,

We would like to thank you for constructive comments and helpful suggestions on the manuscript entitled “Genome-wide analysis of the FBA subfamily of the poplar F-box gene family and its role under drought stress” (ID: 2166385).

Therefore, we have listed below our detailed responses to your comments (except for the suggested language changes, which we have amended accordingly in the revised manuscript). We hope you will find that we have now fully addressed each of your comments.

Reviewer (Previous Decision: minor revisions):

  1. Line 65-72: Introduction: it’s heavy to read, there are too much examples, each described in two rows. I think you should remove some of them, but maybe you can still cite the references.

Response: We agree with this suggestion. Since the Introduction section is a bit cumbersome, we have trimmed it down according to your suggestions.

  1. Line 77: you should revise the order of the references.

Response: We have revised the order of the literature and rechecked the serial numbers of the literature.

  1. Table S6: in this table there are many duplications that you should correct. MYC and MIB are not defined. Other acronims are still not defined.

Response: Thank you for this comment and we totally agree with it! We have annotated all promoter elements and duplicated promoter elements indicate the number of this promoter element for each PtrFBA gene.

  1. Line 303-304: “OE-PtrFBA60 and mutants fba60” - I see in 4.10 construct pBI121-35S:PtrFBA60-GUS and I think that it is OE-PtrFBA60, but I’m not sure. Please use only one name for transgenic plants.

Response: We agree with this suggestion, so we named all the overexpression Arabidopsis in the article OE-PtrFBA60.

  1. Line 467: add what kind of mutant fba60 is.

Response: The fba60 (SALK_019628.33.80.x) mutant was obtained from the Arabidopsis biological re-source center ABRC (ABRC (osu.edu)). The fba60 is a T-DNA insertional mutant.

  1. Finally, please check punctuation throughout the text.

Response: Thank you very much for your suggestions on language and punctuation errors. We have made the changes as you suggested and had someone correct them.

Reviewer 2 Report (Previous Reviewer 2)

This version is much better than the previous one, however, still some sections where the redaction must be improved, mainly, in the materials and methods section. The ideas are unconnected, so it is very difficult to understand the information.

30: Change “What more” to “Moreover”

397-399, This section must be improved, the ideas are not clear.

407, How many bootstraps were used to perform the NJ phylogenetic analysis?

423-437, the redaction of this complete section must be revised, the ideas are not clear. the ideas are disconnected, so is complicated to clearly understand the information.

455-457, The authors said that they obtained the expression data from PopGenIE database. To clarify this section, it is recommended that the authors should add more information about these data, for example, they could include references of transcriptomic data used, type data (RNA-seq or microarrays), if the data downloaded was normalized, etc.

520-523, The Authors described the drought treatment to wild-type and transgenic Arabidopsis plants, however, the information is incomplete. This section must be improved to give more information about abiotic stress treatment. For example, how many days the plants were under drought stress? What means “after the phenotype appeared”? Remember, the material and methods should be enough clear to others can repeat the experiments.

525-529, The statistical analyses section is not clear, authors said that one-way ANOVA and student’s t-test were used to determine the significant differences, but it is not clear when they used one or the other test. After using ANOVA, Did you use some Post hoc tests after applying the ANOVA test? What means that "the data were normalized"?

Author Response

-Dear Reviewer,

We would like to thank you for constructive comments and helpful suggestions on the manuscript entitled “Genome-wide analysis of the FBA subfamily of the poplar F-box gene family and its role under drought stress” (ID: 2166385).

Therefore, we have listed below our detailed responses to your comments. We hope you will find that we have now fully addressed each of your comments.

Reviewer (Previous Decision: minor revisions):

  1. 30: Change “What more” to “Moreover”.

Response: We have made the substitution in the article.

  1. 397-399, This section must be improved, the ideas are not clear.

Response: Thank you very much for your suggestions on the article. We better describe how we obtained the PtrFBAs gene family.

  1. 407, How many bootstraps were used to perform the NJ phylogenetic analysis?

Response: We use the Neighbor-Joining (NJ) method parameter on MEGA7.0 with pairwise deletion and 1000 bootstraps replicates.

  1. 423-437, the redaction of this complete section must be revised, the ideas are not clear. the ideas are disconnected, so is complicated to clearly understand the information.

Response: Thank you for this comment and we totally agree with it! This section is too overwhelming and contains too much non-methodical stuff. We have trimmed and re-narrated it.

  1. 455-457, The authors said that they obtained the expression data from PopGenIE database. To clarify this section, it is recommended that the authors should add more information about these data, for example, they could include references of transcriptomic data used, type data (RNA-seq or microarrays), if the data downloaded was normalized, etc.

Response: We agree with this suggestion, so we referenced that database in the article.

Bhalerao R, Keskitalo J, Sterky F, Erlandsson R, Björkbacka H, Birve SJ, Karlsson J, Gardeström P, Gustafsson P, Lundeberg J, Jansson S. Gene expression in autumn leaves. Plant Physiol. 2003 Feb;131(2):430-42. 

Sundell D, Mannapperuma C, Netotea S, Delhomme N, Lin YC, Sjödin A, Van de Peer Y, Jansson S, Hvidsten TR, Street NR. The Plant Genome Integrative Explorer Resource: PlantGenIE.org. New Phytol. 2015 Dec;208(4):1149-56.

  1. 520-523, The Authors described the drought treatment to wild-type and transgenic Arabidopsis plants, however, the information is incomplete. This section must be improved to give more information about abiotic stress treatment. For example, how many days the plants were under drought stress? What means “after the phenotype appeared”? Remember, the material and methods should be enough clear to others can repeat the experiments.

Response: We agree with your suggestion, so we have added to the section: wild-type and transgenic Arabidopsis seeds were sown simultaneously in soil and grown in a climate-controlled chamber with a light intensity of 180 μ mol/m2/s, a light duration of 14 h /dark duration of 10 h, and a temperature of 25 °C. Arabidopsis were subjected to natural drought treatment for 20 days after one month of growth. Wild-type and mutant Arabidopsis leaves were photographed after water loss and yellowing, and finally treated with rehydration for three days and counted for survival rate and REC.

  1. 525-529, The statistical analyses section is not clear, authors said that one-way ANOVA and student’s t-test were used to determine the significant differences, but it is not clear when they used one or the other test. After using ANOVA, Did you use some Post hoc tests after applying the ANOVA test? What means that "the data were normalized"?

Response: Thank you for this comment and we totally agree with it! We have modified the pictures and descriptions in the article. Statistical analyses were conducted using Microsoft Excel 2016 (Microsoft Corporation, Redmond, WA, USA) and SPSS (version 25, IBM Corporation, Armonk, NY, USA). Before applying the ANOVA test, the data were tested for normality and homogeneity of variance. One-way ANOVA with LSD multiple comparisons test (*P< 0.05, ***P< 0.001.) were performed on the gene relative expression. We made a mistake in our previous statement, Relative expression was measured after normalization with two endogenous reference genes.

Wang HL, Chen J, Tian Q, Wang S, Xia X, Yin W: Identification and validation of reference genes for Populus euphratica gene expression analysis during abiotic stresses by quantitative real-time PCR. Physiol Plant 2014, 152(3):529-545.

Wang HL, Li L, Tang S, Yuan C, Tian Q, Su Y, Li HG, Zhao L, Yin W, Zhao R et al: Evaluation of Appropriate Reference Genes for Reverse Transcription-Quantitative PCR Studies in Different Tissues of a Desert Poplar via Comparision of Different Algorithms. Int J Mol Sci 2015, 16(9):20468-20491.

This manuscript is a resubmission of an earlier submission. The following is a list of the peer review reports and author responses from that submission.

Round 1

Reviewer 1 Report

The paper focus on the FBA subfamily of F-box proteins of poplar. In silico genome wide analysis to evaluate size and features of the family is followed by analysis of expression (by transcriptomics and real-time) of some representative FBAs in stress conditions. Finally, results of under- or over- expression in transgenic plants of a selected interesting component with a regulatory role in plant drought resistance is reported.

The paper is generally well structured but, in my opinion, it has some shortcomings regarding some analyses. The text is often unclear and the methods are not always well described.

Particularly, in results, with regard to transcriptome analysis, it’s not immediately clear which dataset you are referring to. It is explained in some way in Materials and methods, but you give very few details. This makes the reading heavy and uncertain. Also, collinearity is non correctly explained in my opinion. Real-time analysis in the paragraph about “expression patterns” after drought stress should be explicitly mentioned.

In Materials and methods, not all the methods are accurately described as I detail below.

Throughout the text I found a lot of typographical errors and inaccuracy

Please check punctuation throughout the text. It’s sometimes very hard to read it

Line 20: delete “plant”

Line 22: Populus trichocarpa (only the first time, hereafter you can write P. trichocarpa or poplar)

Line 23: Already established (https://doi.org/10.1073/pnas.0812043106), you confirm it

Line 26: pay attention to the use of upper-case letters throughout the text (here In)

Lines 44-45: please change this sentence, I think you meant “Leucine Rich Repeats (LRR) with more than ten conserved domains”

Line 47: please change this sentence adding a full stop after “largest”

Line 47: reference 5 is about Xenopus

Line 53: please move “and 133” before Malus domestica and fit this sentence

Line 59: please fit this sentence

Line 76-78: please fit this sentence, I can’t understand

Line 81: please add a full stop after “analysis” and remove “and”

Line 81: the tree shown in Supplementary Figure S2 is unreadable, please remove or modify it

Line 101: please add something, maybe a legend or a reference or a file in supplementaries, to explain the meaning of these acronyms

Line 81: please fit this sentence, maybe remove “The” before number

Lines 109-111: please fit this sentence moving “from the poplar FBA family” after PtrFBA60

Figure 2b: you say in the text that the number of F-box genes in P. trichocarpa is 337

Lines 129-131: please move this sentence in methods

Line 136: please fit the sentence “located in 6 on chromosome number”

Line 137: please explain in “Methods” how you did it

Line 140: please explain “WGD pair”

Lines 141-142: please fit the sentence, maybe “The Ka/Ks values of PtrFBA7/PtrFBA68 and PtrFBA65/PtrFBA25 pairs were…”

Figure 4: in caption you call “collinear” gene pairs something that maybe should be called “paralogs”. I don’t agree with your interpretation. Can you explain it or fit it?

Line 150: please move in “Methods” the reference to MCScanX

Line 163: change May with may

Line 165: how can you assert “high collinearity with woody apples”, I see the presence of orthologous genes but not collinearity from your data

Table S4: it’s incomplete, please fit it

Line 214: STRE is not present in Table S4, MYC and MIB are not defined

Figure 7: the colours of the bars under a) from the colours in b). Please fit it

Line 250: to help reading specify that you used a Quantitative real-time (RT-qPCR) analysis

Line 252: “the expression of PtrFBA6/11/13/50 was down-regulated” I think this effect is stronger in PtrFBA36 than in PtrFBA50

Line 271: RWC is REC in caption of Figure 11c. Fit it and explain what this acronym stays for

Line 297: remove one “distributed”

Line 297: remove “and”

Lines 307-308: could you please explain better, “this change” is not clear for me

Line 309: rewrite the sentence “segmented duplication genes and 12 tandem repeat genomes”

Line 318: please cite Figure 2b

Line 321: please fit the sentence “Plant apple has a higher substitution rate and a closer genetic relationship”. Also, what do you refer to with “ higher substitution rate”?

Line 338: correct ERE with ARE

Line 354: please delete “was” to fit the sentence “the prediction was confirmed the PtrFBAs protein sequences”

Line 357: fit this: P. trichosanthes 4.1 genome

Lines 366-368: delete the sentence “Predicting cis-acting elements of promoters using PlantCARE” and move the URL after “PlantCARE” on line 368

Lines 391-392: “Dark-processed and dehydrated transcriptome data were obtained from laboratory transcriptome sequencing results” you should detail something about this sequencing, how did you do the sequencing? is there a public dataset, in GEO for example, or other references?

Line 406: Delete “(Supplementary Table S4)” which is not correct and is cited again correctly after

Line 409: Normalization is not well explained, normally it is performed using the expression of a reference gene in addition to a control. How did you do it?

Line 410: Add some particular al least on Master Mix used

Line 417: delete “to put it simply” and fit the protocol in a formal way – what is the cling film?

Reviewer 2 Report

The manuscript contains too many errors (typing and redaction) that greatly complicate smooth reading. Likewise, several figures are difficult to interpret because the quality is poor and/or because not enough information is included to understand the results shown.

In the PDF file, I point out several of the errors found and sections with poor writing that complicate the reading of the manuscript.

In general, the manuscript quality must be improved. The material and methods are poor, too much information was omitted, which difficulted to understand the results and conclusions claimed by the authors. This section must be improved.

There are too many sections that need to be improved, here only listed a few of them.

90-91: What is the difference between removing repetitive sequences and redundant sequences?

It is impossible to see the information in figure S2, what are the most relevant results of the Evolutionary developmental analysis?

107-108, 112-113: Should give more information about conserved residues.

Figure 2b. The number of F-box proteins (338) does not coincide with those described in the results section by P. trichocarpa.

138-141, this section is not clear. It must be clarified.

144-146, this section needs to clarify, it is not clear where come from these “13 pairs of homologous genes”.

147-149, Which is the evidence of these twelve genes were duplicated in block "fragment"?

161-163, the authors need to show more evidence to support this observation.

Check the quality and order of figures, and tables. It does not correspond to the order cited in the manuscript.

Fig 6a, it is very difficult to see the information. Figure S4 was not cited in the manuscript.

215-218, Which is the evidence of the cis-elements (ABA response and LTR) were enriched in the promoters of PtrFBAs?

Figure 8. What values ​​are indicated on the scale of figures 8a and 8b? Explain why you said the transcriptome expression is considered as "relative expression values…" Why were these genes chosen to evaluate their response to drought?

263-265, it is difficult to follow this section, it should be improved.

Figure 10, which is the evidence of the PtrFBA60 is located on “small vesicles? This figure says that results were obtained from tobacco plants, but the methods section doesn't describe it.

super::GFP and Super::PtrFBA60 constructions must be explained in the methods section.

271, What means RWC?

Figure S7, the description of this figure does not correspond to the information shown.
